# A second open reading frame in human enterovirus determines viral replication in intestinal epithelial cells

Haoran Guo[1,2], Yan Li[2], Guanchen Liu[2], Yunhe Jiang[3], Siyu Shen[2], Ran Bi[1], Honglan Huang[3], Tong Cheng[4], Chunxi Wang[1] & Wei Wei[1,2]

Human enteroviruses (HEVs) of the family *Picornaviridae*, which comprises non-enveloped RNA viruses, are ubiquitous worldwide. The majority of EV proteins are derived from viral polyproteins encoded by a single open reading frame (ORF). Here, we characterize a second ORF in HEVs that is crucial for viral intestinal infection. Disruption of ORF2p expression decreases the replication capacity of EV-A71 in human intestinal epithelial cells (IECs). Ectopic expression of ORF2p proteins derived from diverse enteric enteroviruses sensitizes intestinal cells to the replication of ORF2p-defective EV-A71 and respiratory enterovirus EV-D68. We show that the highly conserved WIGHPV domain of ORF2p is important for ORF2p-dependent viral intestinal infection. ORF2p expression is required for EV-A71 particle release from IECs and can support productive EV-D68 infection in IECs by facilitating virus release. Our results indicate that ORF2p is a determining factor for enteric enterovirus replication in IECs.

[1] Key Laboratory of Organ Regeneration and Transplantation of Ministry of Education, Institute of Translational Medicine, First Hospital, Jilin University, Changchun, Jilin 130021, China. [2] Institute of Virology and AIDS Research, First Hospital, Jilin University, Changchun, Jilin 130021, China. [3] Department of Pathogen Biology, The Key Laboratory of Zoonosis, Chinese Ministry of Education, College of Basic Medical Science, Jilin University, Changchun, Jilin 130021, China. [4] State Key Laboratory of Molecular Vaccinology and Molecular Diagnostics, School of Public Health, Xiamen University, Xiamen, China. Correspondence and requests for materials should be addressed to W.W. (email: wwei6@jlu.edu.cn)

Enterovirus, a genus in the family Picornaviridae, consists of 13 species, and 7 of these species contain human pathogens. Human enteroviruses (HEVs), including poliovirus, coxsackieviruses, echoviruses, rhinoviruses and the subgroups enteroviruses, are ubiquitous worldwide[1,2]. These diverse enteroviruses cause a diverse array of clinical features, including aseptic meningitis; hand, foot, and mouth disease; neonatal sepsis-like disease; pancreatitis; encephalitis; myocarditis and pericarditis; paralysis and respiratory diseases[3–8]. Enteroviruses are transmitted predominantly by the faecal–oral route or by respiratory droplets[9–11]. Enteroviruses enter the host via the oral cavity or respiratory tract. The primary sites of replication are presumed to be the gastrointestinal and respiratory epithelium, from where these viruses can disseminate to the target organs via the blood circulation.

HEVs are small, non-enveloped icosahedral viruses. The genome of HEVs is a positive-sense single stranded RNA genome, including a long single open reading frame (ORF) flanked by a 5′ untranslated region (UTR) and 3′ UTR. It has been well established that HEV proteins are derived from viral polyproteins encoded by a single ORF[12–14]. This polyprotein is further hydrolyzed to form 4 structural proteins (VP1, VP2, VP3, and VP4) and 7 non-structural proteins (2A, 2B, 2C, 3A, 3B, 3C, and 3D)[9].

In the present study, we characterize a second ORF in the enterovirus genome and its encoded protein ORF2p. ORF2p-defective EV-A71 has decreased viral infectivity in both transformed and freshly isolated primary human IECs. Diverse enterovirus ORF2p proteins show an ability to sensitize human intestinal HT-29 cells to the replication of ORF2p-defective EV-A71 and respiratory enterovirus EV-D68. Mutation of the conserved WIGHPV domain of ORF2p destroys its intestinal infection capacity, and ORF2p enhances the viral replication capability of EV-A71 and EV-D68 by facilitating virus release from gut cells. Our findings shed light on an determinant for enterovirus replication at the primary site of replication, namely, intestinal cells.

## Results

**EV-A71 ORF2 is efficiently translated in infected cells.** We previously screened the susceptibilities of different cell lines to an enteric enterovirus (EV-A71) and a respiratory enterovirus (EV-D68). This approach enabled us to identify ICAM-5/telencephalin as a functional receptor on the basis of its differential expression in permissive and non-permissive cells; this receptor impacts the entry step of the EV-D68 life cycle but has no effect on EV-A71 entry[15]. In addition, we noticed that HT-29 and LS 174 T human IECs were susceptible to EV-A71 but not to EV-D68 (Supplementary Fig. 1a–c). Specifically, the RNA levels of progeny EV-D68 virus in the culture medium were dramatically lower than those of EV-A71; in contrast, little difference in intracellular capsid protein (VP1) expression and RNA replication was found between EV-A71 and EV-D68. These findings suggest that all preceding steps in EV-D68 replication were unaffected and that virus release from HT-29 cells was blocked (Supplementary Fig. 1d–f).

To identify the viral molecules required for EV-A71 replication in HT-29 cells, we performed functional analysis of different chimaeras of EV-A71 and EV-D68. Surprisingly, our data indicated the 5′ UTR to be a determinant of EV-A71 replication in HT-29 cells (Supplementary Fig. 1g). The 5′ UTR of enteroviruses is highly structured and directs viral RNA synthesis and viral mRNA translation[16,17]. Our study suggested a role for the 5′ UTR in EV-A71 release from human intestinal cells, and we hypothesized that the EV-A71 5′ UTR may encode an

unknown protein that is critical for intestinal virus replication. There are eight AUG codons in all three reading frames of the EV-A71 strain AH08/06 5′ UTR, and the $AUG_{589}$ triplet located in the VI domain of the viral internal ribosomal entry site (IRES) is highly conserved in HEVs but not in EV-D68 (Supplementary Fig. 2a). According to previous studies, the corresponding triplet in poliovirus can function as an initiation codon under artificial conditions[16,18]. The ORF initiated by $AUG_{589}$ encodes a 71-amino acid polypeptide that we named ORF2p (Fig. 1a, b). We inserted a haemagglutinin (HA) tag into the ORF2p sequence of the EV-A71 infectious clone and subsequently transfected the in vitro-synthesized RNA transcripts into HT-29 cells (Supplementary Fig. 2b). Interestingly, a protein of ~10 kDa was detected in the lysates of treated cells using an anti-HA antibody (Supplementary Fig. 2c). To determine whether $AUG_{589}$ is active in EV-A71-infected intestinal cells, we generated antiserum by immunizing rabbits with the polypeptide derived from ORF2p and purified the antibody using an ORF2p antigen column. Immunoblotting also revealed the protein in the lysates of EV-A71-infected HT-29 cells and primary IECs (Fig. 1c). The viral specificity of ORF2p was confirmed by its absence in HT-29 and IEC cells infected by a mutant virus (EV-A71ΔORF2p) with a truncated ORF2p (Fig. 1c).

**ORF2p facilitates intestinal infection by EV-A71.** An alternative *orf* of EV-A71 was sufficiently translated in viral-infected intestinal cells, and we thus investigated the replication capacity of wild-type and ORF2p-deficient EV-A71 in different cells to address the contribution of ORF2p to viral replication. EV-A71ΔORF2p was able to grow in RD cells as efficiently as did wild-type EV-A71 (Fig. 2a); however, compared the wild-type virus, the infectivity of the virus was significantly decreased in HT-29 and primary IEC cells (Fig. 2b, c). Ectopic expression of EV-A71 ORF2p restored the sensitivity of HT-29 cells to EV-A71ΔORF2p but had only a modest effect on wild-type EV-A71 virus infection (Fig. 2d). When we examined EV-A71 replication in a panel of intestinal epithelial cell lines, we found that deletion of orf2p caused a marked reduction of virus titers in several cell lines, and smaller, but still significant reductions in the others (Supplementary Fig. 3a). In addition, the essential role of ORF2p in facilitating EV-A71 replication is cell-type dependent and not only in intestinal cell lines but also in the mouse neuron NSC-34 cells (Supplementary Fig. 3b). EV-A71 can be phylogenetically classified into three genogroups (A, B, and C), which include 12 genotypes (A, B0–B5, and C1–C5). ORF2p is highly conserved among all these genotypes (Supplementary Fig. 4), and our results indicate that ORF2p variants from different EV-A71 subtypes might render HT-29 cells susceptible to EV-A71ΔORF2p replication (Fig. 2e). These findings demonstrate that ORF2p plays a substantial role in enterovirus replication in IECs.

Enteroviruses spread through the faecal–oral route or via respiratory transmission[9–11] and can disseminate from their primary sites of replication in the gastrointestinal or respiratory tract to infect other tissues and organs[19]. For most human enteric A, B and C enteroviruses, ORF2 that begins at the 3′ border of the 5′UTR encodes a highly conserved polypeptide; this sequence is absent in human respiratory enteroviruses, including EV-D68 and rhinovirus. The specific retention of *orf2* in the majority of enteric enteroviruses hints at a conserved function, which may involve regulation of viral replication capacity in human enteric cells. Thus, derivatives of HT-29 cells stably expressing ORF2p from enterovirus A (CV-A16 and EV-A71), enterovirus B (CV-B3, Echovirus 6, Echovirus 19, and EV-B73) and enterovirus C (poliovirus 1, CV-A24, and EV-C96) were established via retroviral vector transduction. These cells were challenged with

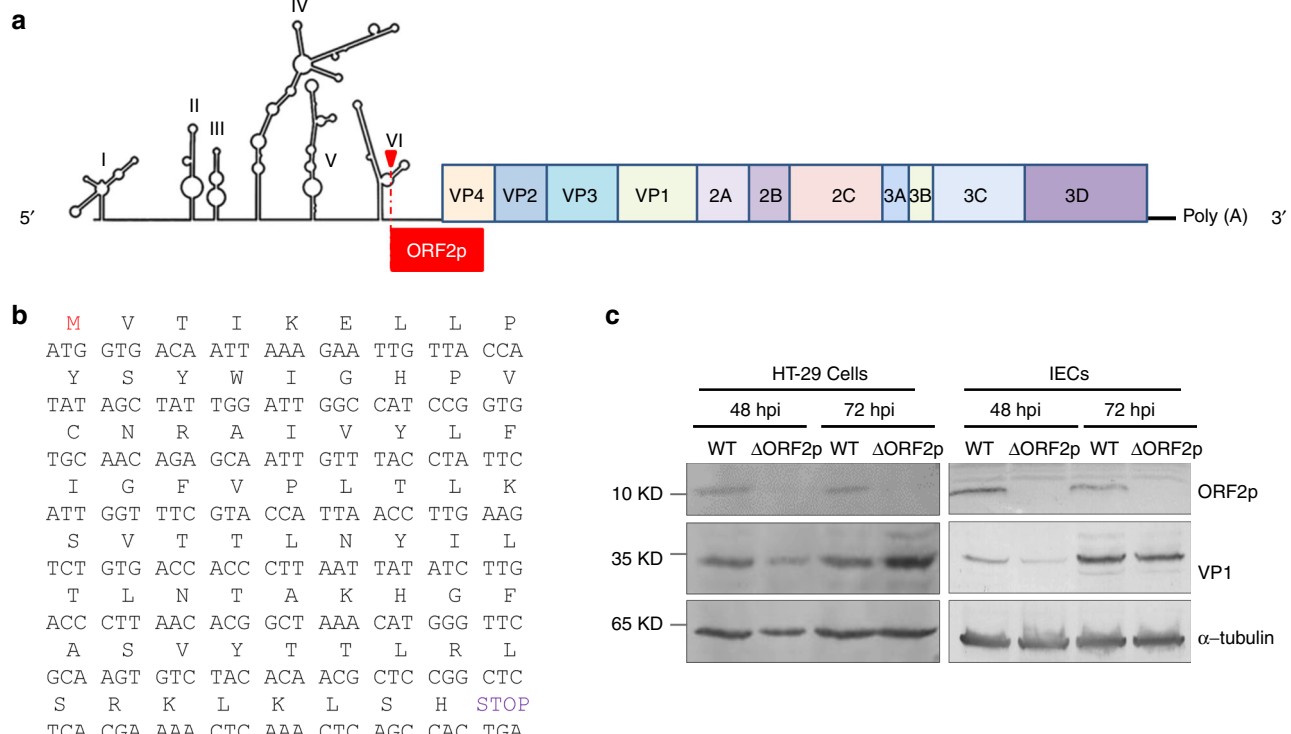

**Fig. 1** A second ORF of EV-A71 is efficiently translated in challenged cells. **a** Schematic diagram of the EV-A71 structure. The EV-A71 genome contains a long ORF flanked by a 5'UTR and 3'UTR. The ORF encodes a 250-kDa polyprotein that is processed into the P1, P2 and P3 regions, which are further cleaved into mature proteins (VP1 to VP4, 2 A to 2 C, and 3 A to 3D). Red box, the second ORF, which begins at the 3' border of the IRES in the 5'UTR, encodes a 64-76-amino acid polypeptide in diverse sub-genotypes of EV-A71. **b** Amino acid and nucleic acid sequences of ORF2p from EV-A71 strain AH08/06. **c** HT-29 cells and isolated primary IECs were infected with EV-A71 or EV-A71ΔORF2p at an MOI of 0.1. Cells were harvested and prepared for immunoblotting at 48 hpi and 72 hpi using antibodies against EV-A71 ORF2p, VP1 and α-tubulin. Source data are provided as a Source Data file

EV-A71ΔORF2p at a multiplicity of infection (MOI) of 0.1, and the titres of progeny viruses in the culture medium were determined at 72 h post-infection (hpi). The data indicated that ectopic expression of different ORF2p variants conferred efficient replication of ORF2p-deficient EV-A71 in intestinal HT-29 cells (Fig. 3a). We further demonstrated that ORF2p from enteric enteroviruses confers replication capacity on respiratory EV-D68 in HT-29 cells (Supplementary Fig. 5).

**The WIGHPV domain of ORF2p is important for viral infection.** Next, we evaluated whether specific domains of ORF2p are important for its ability to facilitate viral replication in intestinal cells. Comparison of the amino acid sequences of ORF2p proteins showed that their N-terminal regions share high similarities and contain a WIGHPV domain that is highly conserved among ORF2p proteins from different species of HEVs (Fig. 3b). To assess whether the WIGHPV domain is essential for ORF2p function, we utilized alanine-scanning mutants of ORF2 by generating two mutated EV-A71 viruses carrying the ORF2p mutants WIG/AAA and HPV/AAA and investigated their growth kinetics in HT-29 cells. Although these cells partially supported viral replication of the mutated viruses, the final viral titres of EV-A71 (ORF2p WIG/AAA) and EV-A71 (ORF2p HPV/AAA) were approximately 0.005% and 1.75% of that of the wild-type virus, respectively (Fig. 3c). Consistent with this finding, compared to expression of the wild-type ORF2p protein, ectopic expression of ORF2p WIG/AAA or HPV/AAA dramatically impaired the ability of the protein to support EV-A71ΔORF2p replication in HT-29 cells (Fig. 3d).

**ORF2p enhances the release of enterovirus particles.** To test the notion that EV-A71 ORF2p contributes to overcoming the specific restriction for virus release from IECs, we first confirmed the lack of a distinct difference in virus attachment, endocytosis, 5'UTR activity and pseudovirus infectivity between EV-A71 and EV-A71ΔORF2p in HT-29 cells (Supplementary Fig. 6). Next, we transfected isolated primary IECs with equal amounts of in vitro-synthesized RNA transcripts for EV-A71 or EV-A71ΔORF2p. Time-dose analysis indicated strongly decreased viral RNA accumulation in the supernatant of EV-A71ΔORF2p RNA-transfected IECs compared to that of EV-A71 RNA-transfected cells (Fig. 4a), even though equal amounts of EV-A71 or EV-A71ΔORF2p viral RNA were detected in the cell lysates (Fig. 4b). Furthermore, we noticed that the levels of intracellular vRNA of EV-A71ΔORF2p were modestly higher than those of EV-A71 vRNA at 8 and 10 h post-transfection (Fig. 4b). This phenomenon may be due to the deficiency in viral RNA release in the absence of ORF2p expression, which could lead to higher rates of intracellular vRNA accumulation compared to EV-A71. Overall, markedly fewer infectious virions were detected in the supernatant of IECs transfected with EV-A71ΔORF2p RNA, as based on a virus titre assay (Fig. 4c). However, similar amounts of EV-A71 and EV-A71ΔORF2p infectious particles were released when the samples (both cells and supernatant) were subjected to three cycles of freeze–thawing (Fig. 4c). Additionally, one-step growth curve analysis of primary IECs or HT-29 cells infected with high doses (MOI = 10) of EV-A71 or EV-A71ΔORF2p showed that the titre of extracellular EV-A71ΔORF2p was dramatically lower than that of extracellular EV-A71, though equal amounts of infectious virions of intracellular EV-A71 or EV-A71ΔORF2p

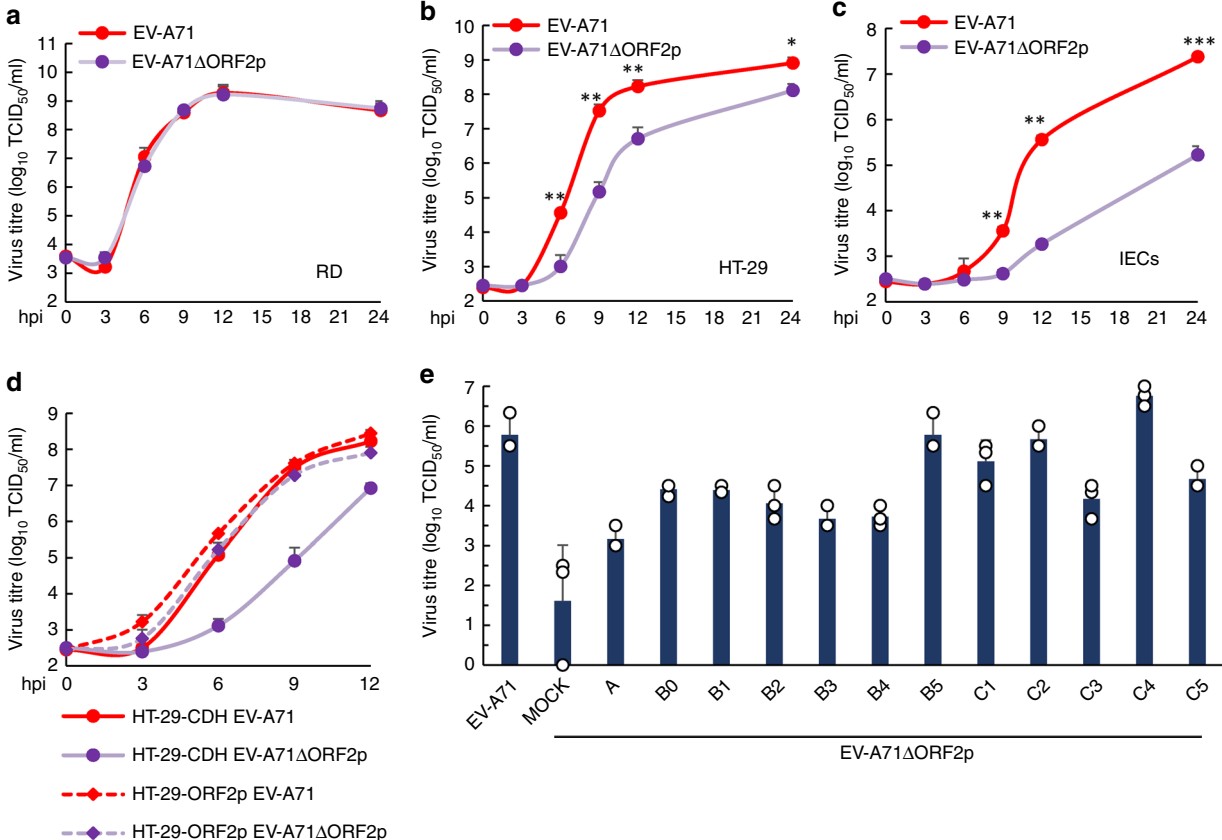

**Fig. 2** ORF2p is crucial for intestinal infection by EV-A71. **a–c** One-step growth curve of EV-A71 and EV-A71ΔORF2p in RD, HT-29 and isolated primary IECs at an MOI of 10. Error bars denote SEM; ANOVA test, $n = 3$ biologically independent experiments; *$p < 0.05$, **$p < 0.01$, ***$p < 0.001$. **d** HT-29 cells stably expressing EV-A71 ORF2p or control HT-29-CDH cells generated by transduction with the empty lentiviral vector pCDH-CMV-MCS-EF1-Puro were challenged with EV-A71 WT or EV-A71ΔORF2p at an MOI of 10. Viral titres were determined at 0, 3, 6, 9, and 12 hpi ($n = 3$ biologically independent experiments). Error bars denote SEM. **e** HT-29 cells expressing the ORF2p protein derived from EV-A71 sub-genotypes A, B0, B1, B2, B3, B4, B5, C1, C2, C3, C4 and C5 were infected with EV-A71ΔORF2p at an MOI of 0.1. Viral titres were determined at 72 hpi ($n = 3$ biologically independent experiments). Error bars denote SEM. Source data are provided as a Source Data file

were detected in the cell lysates (Fig. 4d, Supplementary Fig. 7a). Consistent with this, ORF2p specifically facilitated accumulation of the EV-A71 structural protein VP1 in the supernatant (Fig. 4e). In a control experiment, there was no significant difference between the titre of EV-A71 or EV-A71ΔORF2p in RD cells (Supplementary Fig. 7b). Importantly, ectopic expression of EV-A71 ORF2p notably relieved the inhibition of EV-A71ΔORF2p particle release from HT-29 cells and had a mild effect on intracellular VP1 expression (Fig. 4f).

Our findings led us to investigate whether EV-A71 ORF2p can facilitate EV-D68 release from HT-29 cells. ORF2p expression resulted in marked cytopathic effects in EV-D68-infected cells (Fig. 4g). Again, the release of EV-D68 particles was substantially enhanced in the presence of ORF2p during viral replication in HT-29 cells (Fig. 4h), whereas intracellular VP1 levels were marginally affected by expression of ORF2p (Fig. 4i). All these results show a direct role for the enterovirus ORF2p protein in facilitating enterovirus release from IECs.

Analysis of the amino acid sequence of ORF2p using SMART software[20] revealed a consensus transmembrane region that is conserved among human enteric enteroviruses (Supplementary Fig. 8a). Mutation of three conserved residues (Y25, G29, P32) in the transmembrane region impaired the ability of ORF2p to support EV-A71ΔORF2p and EV-D68 replication in HT-29 cells (Supplementary Fig. 8b). Interestingly, immunofluorescence assays showed ORF2p to be almost exclusively located in

the cytoplasm of ORF2p-overexpressing cells, and ORF2p wild-type but not ORF2p YGP25/29/32AAA proteins notably accumulated in cytoplasmic vesicles (Supplementary Fig. 8c). Moreover, the number of eGFP-LC3 puncta (a marker of autophagosomes) was dramatically increased in ORF2p-transfected HT-29 cells (Supplementary Fig. 9a, b) and strongly colocalized with mCherry-tagged EV-A71 ORF2p by confocal microscopy (Supplementary Fig. 9a, c). In addition, treatment with the autophagy inhibitor wortmannin impaired ORF2p localization to cytoplasmic vesicles (Supplementary Fig. 9d). Our results further indicated that a lack of ORF2p expression suppressed EV-A71-induced conversion of LC3-I to LC3-II in HT-29 cells but had no detectable influence in HEK293T cells (Supplementary Fig. 9e). These findings highlight a positive relationship between EV-A71 ORF2p expression and the host autophagic response. Intracellular vesicle transport is exploited by diverse viruses[21], participating in enterovirus release from human IECs[22], and autophagy is activated during EV-A71 infection to facilitate viral replication *in vitro* and *in vivo*[23–26]. Our findings suggest that ORF2p facilitates virus release by manipulating the intracellular vesicle transport pathway, and further investigation is warranted. It should also be noted that disrupting ORF2p expression markedly increased survival rates in a lethal EV-A71-challenged neonatal mouse model, revealing an essential role for ORF2p in EV-A71 virulence *in vivo* (Supplementary Fig. 10).

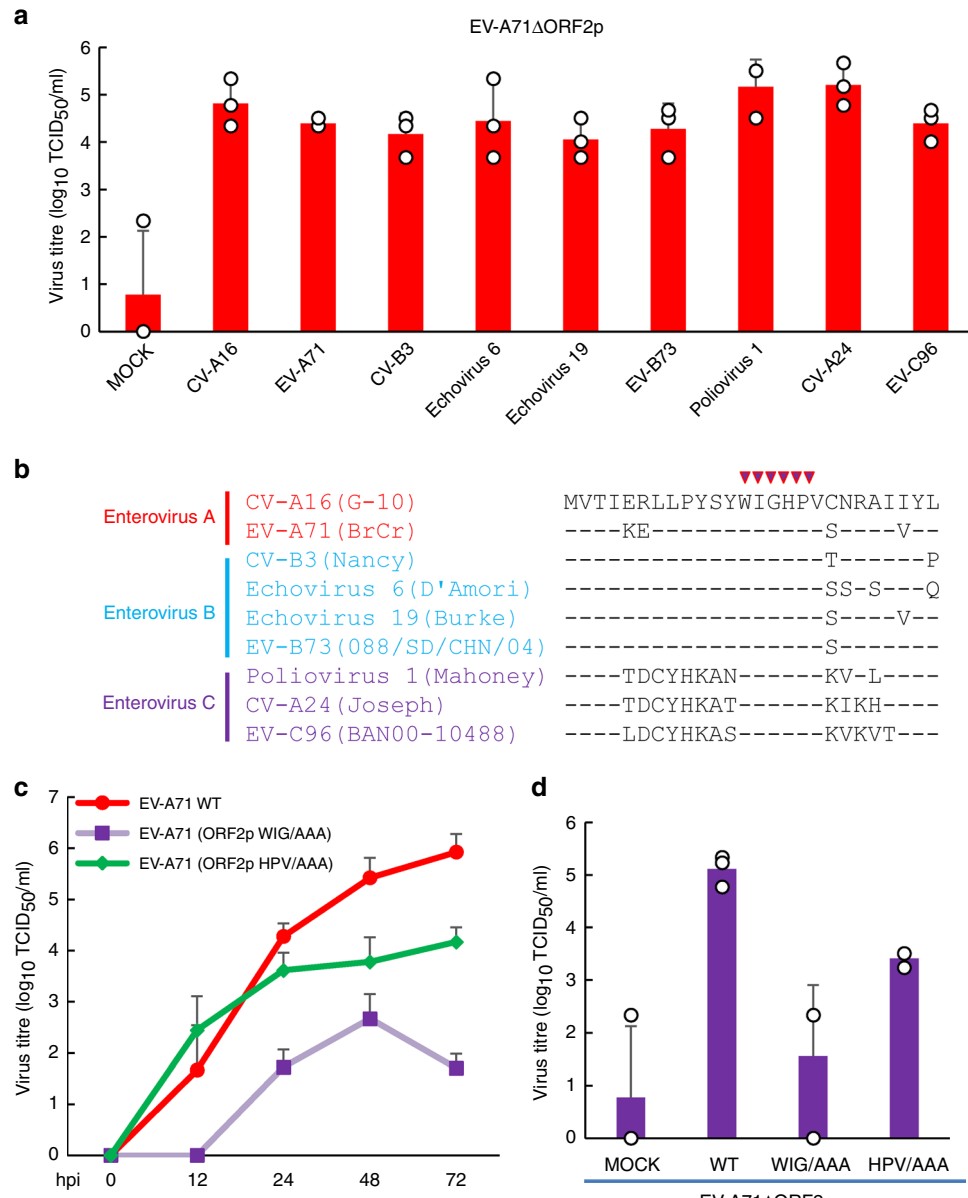

**Fig. 3** The WIGHPV domain of ORF2p from enteric enteroviruses is important for viral infection in HT-29 cells. **a** HT-29 cells expressing the ORF2p protein derived from CV-A16, EV-A71, CV-B3, Echovirus 6, Echovirus 19, EV-B73, Poliovirus 1, CV-A24 and EV-C96 were infected with EV-A71ΔORF2p at an MOI of 0.1. Viral titres were determined at 72 hpi ($n = 3$ biologically independent experiments). Error bars denote SEM. **b** Alignment of amino acid sequences of the N-terminal portion of ORF2p from human enteroviruses. Residues of WIGHPV are marked with red arrows. **c** HT-29 cells were challenged with EV-A71 and mutant viruses EV-A71 (ORF2p WIG/AAA) and EV-A71 (ORF2p HPV/AAA) at an MOI of 0.1. Viral titres were determined at 0, 12, 24, 48, and 72 hpi ($n = 3$ biologically independent experiments). Error bars denote SEM. **d** Viral titres of HT-29 cells expressing wild-type ORF2p, WIG/AAA, or HPV/AAA; control cells were infected with EV-A71ΔORF2p. Viral titres were determined at 72 hpi. Error bars denote SEM, $n = 3$ biologically independent experiments. Source data are provided as a Source Data file

## Discussion

For decades, the human enterovirus genome was thought to contain a single open reading frame plus two non-coding regions (5′UTR and 3′UTR). In the present study, we characterized a second ORF in the EV-A71 genome that is efficiently translated in infected cells. ORF2, initiated by $AUG^{589}$, is mainly expressed from the 5′ UTR and overlaps with the VP4 sequence of the main ORF. The amino acid sequence of the ORF2 product ORF2p is highly conserved among diverse HEVs (subtypes A, B, and C), suggesting an important role for ORF2p in HEV replication or transmission. In accordance with our findings, Lulla et al. recently

identified the upstream open reading frame in the viral genomes of echovirus 7 and poliovirus 1[27].

ORF2p is required for the full replication capacity of EV-A71 in intestinal cell lines and isolated primary IECs. However, defects in ORF2p do not observably influence EV-A71 attachment to the host cell membrane, endocytosis, 5′UTR activity, viral RNA replication or polyprotein translation (Supplementary Fig. 6), demonstrating that ORF2p has mild effects at the early steps of the EV-A71 life cycle. In contrast, ORF2p is responsible for infectious particle release from EV-A17-infected or viral RNA-transfected IECs (Fig. 4a–d). Overall, ORF2p is a enterovirus

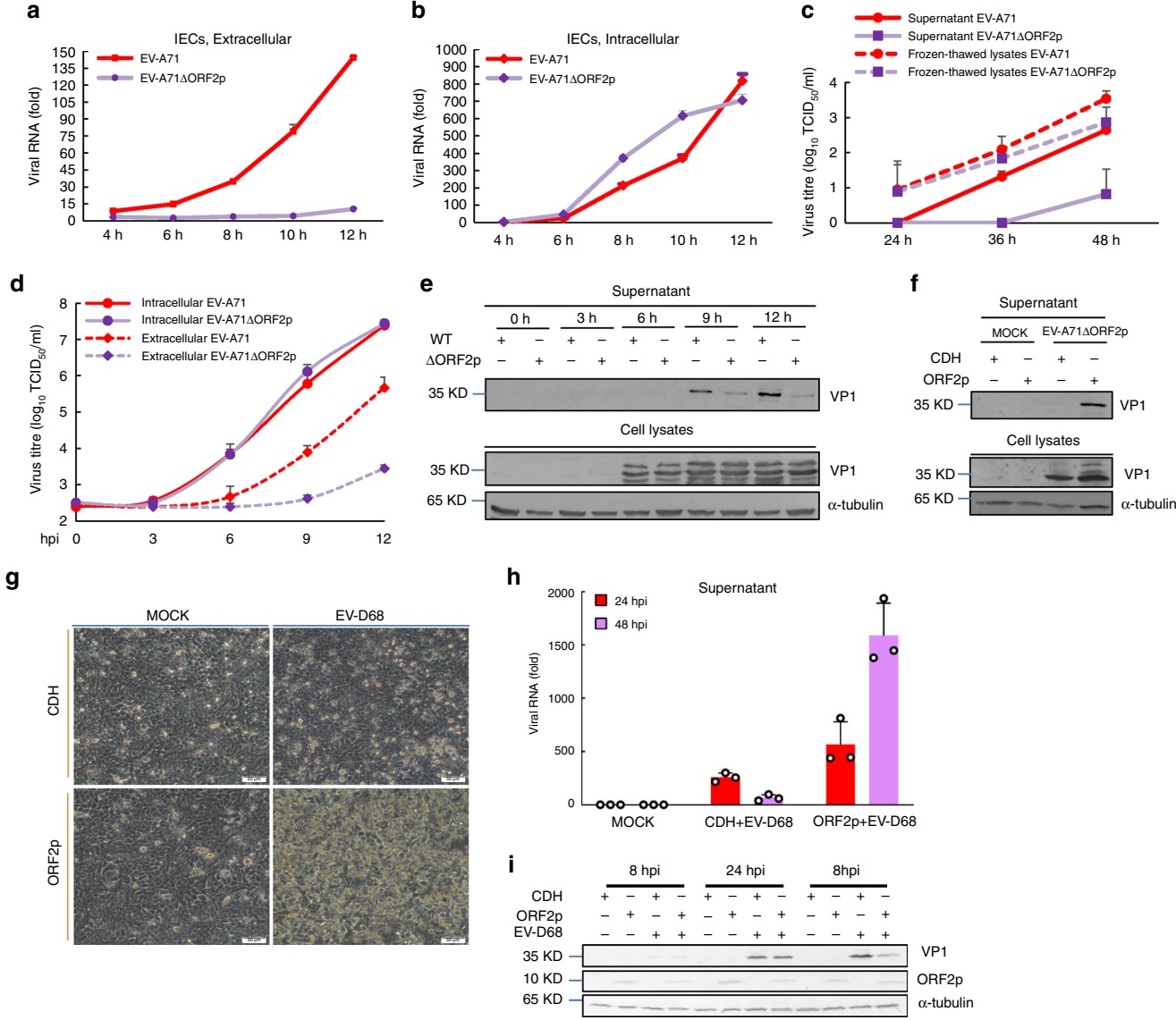

**Fig. 4** ORF2p facilitates the release of enterovirus particles from isolated IECs and HT-29 cells. **a–c** Isolated primary IECs were transfected with 1 μg *in vitro*-synthesized RNA transcripts of EV-A71 or EV-A71ΔORF2p (*n*=3 biologically independent experiments). Error bars denote SEM. Viral RNA in the culture supernatant **a** and cell lysates **b** was detected by an RT-PCR assay at the indicated times. **c** Virus titres of the supernatant or frozen-thawed lysates were determined at 24, 36, 48 h post-transfection. **d, e** Isolated primary IECs were challenged with EV-A71 or EV-A71ΔORF2p at an MOI of 10. **d** Virus titres of EV-A71 or EV-A71ΔORF2p in the supernatant or cell lysates were determined at the indicated times (*n*=3 biologically independent experiments). Error bars denote SEM. **e** The supernatant and cells were harvested at indicated time, followed by immunoblot analysis, as described above. **f** HT-29 cells stably expressing EV-A71 ORF2p or control HT-29-CDH cells were challenged with EV-A71ΔORF2p at an MOI of 0.1. The supernatants and cells were harvested for immunoblotting using antibodies against EV-A71 VP1 and α-tubulin. CDH, control HT-29-CDH cells. **g** Detection of the EV-D68-induced cytopathic effect (CPE). HT-29-ORF2p or control HT-29-CDH cells were infected with equal amounts of EV-D68, and CPE was imaged via light microscopy. Scale bars equal 50 μm. **h, i** HT-29-ORF2p or control HT-29-CDH cells were infected with equal amounts of EV-D68. **h** The supernatant was harvested at 24 and 48 hpi for RT-PCR assays (*n*=3 biologically independent experiments). Error bars denote SEM. **i** Cells were harvested at 8, 24, and 48 hpi, followed by immunoblot analysis. Source data are provided as a Source Data file

virulence effector that plays a critical role at the late egress stage during virus replication in IECs.

Picornaviruses have a short cytosolic lifecycle that ends in cell lysis. However, recent studies have reported that these non-enveloped RNA viruses subvert and short-circuit macro-autophagy/autophagy-related pathways for virus particle release[28,29]. In this study, we identified that EV-A71 ORF2p is a membrane-associated protein that is localized to cytosolic vesicles. The inactive mutants that disrupt ORF2p localization to vesicles correlate with defects in ORF2p-dependent virus replication (Supplementary Fig. 8). Interestingly, ORF2p is associated with EV-A71 infection-induced autophagy activation (Supplementary Fig. 9); ectopically expressed ORF2p co-localized with LC3-GFP and was sufficient for autophagy signalling activation. All these findings suggest a possible relationship between ORF2p-triggered autophagy and the enhancement of viral intestinal infection by ORF2p. The autophagy pathway is known to be important for enterovirus replication at different steps in many cell types, and future studies are needed to further understand the mechanisms and functions of ORF2p-induced autophagy.

Enteroviruses are frequently encountered human pathogens worldwide[30–34]. Although infection by these viruses provokes

acute disease, a role for enterovirus infection in chronic diseases, such as Parkinson's disease, amyotrophic lateral sclerosis and type 1 diabetes mellitus, is suspected[35–37]. Similar to many RNA viruses, enteroviruses have great genetic variability caused by point mutations and recombination. The primary replication site of most enteroviruses is the gastrointestinal tract[19], and the identification of ORF2p as a determinant for the full replication capacity of HEVs is not only helpful for a better understanding of enterovirus pathogenesis and transmission, but this protein may also constitute a promising target for the discovery of anti-HEV drugs.

Determinants of the neurovirulence of other enteroviruses such as poliovirus[38–42] and the cardiovirulence of CV-B1[43] and CV-B3[44,45] are closely related to 5'UTR sequences. Our data showed that expression of ORF2p reduces survival rates in a lethal EV-A71-challenged neonatal mouse model but that this was not strongly related to intestinal virus replication, which leads us to speculate a possible role for ORF2 in virulence and tropism for other tissues. Clarifying the potential effects of ORF2p on the tropisms of other enteroviruses is still necessary.

Our findings, in agreement with other recent work[27], show that HEVs harbour a second ORF that encodes the ORF2p protein to support viral intestinal infection. The purposeful inhibition of ORF2p translation and function may represent an attractive pharmacological intervention against HEVs.

## Methods

**Plasmids and reagents**. The plasmids pEGFP-LC3 (Addgene, 24920), pRSV-Rev (Addgene, 12253), pMDLg/pRRE (Addgene, 12251), and pCMV-VSV-G (Addgene, 8454) were purchased from Addgene. EV-A71 BrCr/USA/1970 (Gen-Bank: U22521), EV-A71 10857/NED/1966 (GenBank: AB575912), EV-A71 11977/NED/1971 (GenBank: AB575913), EV-A71 20233/NED/1983 (GenBank: AB575923), EV-A71 MY821-3/1997 (GenBank: DQ341367), EV-A71 5865/sin/000009/SIN/2000 (GenBank: AF316321), EV-A71 5511-SIN-00 (GenBank: DQ341364), EV-A71 5511-SIN-00 (GenBank: DQ341364), EV-A71 NED/1991 (GenBank: AB575935), EV-A71 Tainan/5746/98/TW/1998 (GenBank: AF304457), EV-A71 06-KOR-00/KOR/2000 (GenBank: DQ341355), EV-A71 SHZH98/CHN/1998 (GenBank: AF302996), EV-A71 2007-07364/TW/2007 (GenBank: EU527983), CV-A16 G-10 (GenBank: U05876.1), CV-B3 Nancy (GenBank: JX312064.1), Echovirus 6 D'Amori (GenBank: AY302558.1), Echovirus 19 Burke (GenBank: AY302544.1), EV-B73 088/SD/CHN/04 (GenBank: KF874626.1), poliovirus 1 Mahoney (GenBank: V01149.1), CV-A24 Joseph (GenBank: EF026081.1), and EV-C96 BAN00-10488 (GenBank: EF015886.1) ORF2p expression vectors were obtained from Generay Biotech Co. Ltd. (Shanghai, China). In brief, fragments containing the coding sequences of ORF2p variants flanked by 5′ EcoR I sites and 3′BamH I sites were inserted into the pCDH-CMV-MCS-EF1-Puro vector (System Biosciences, LLC). pCDH-CMV-MCS-EF1-Puro-ORF2p mutants used in this study were obtained by single-site mutation. pEV-A71 ORF2p-HA was amplified by PCR and cloned into the VR1012 vector. An infectious cDNA clone of pA12-EV-A71 (AH08/06) was kindly provided by Dr. T. Cheng. EV-A71ΔORF2p mutants were generated by using a Q5® Site-Directed Mutagenesis Kit (New England Biolabs) to introduce a stop codon at residue 6 of ORF2p. EV-A71 (OR-HA-F2), EV-A71 WIG/AAA and EV-A71 HPV/AAA were also generated by site-specific mutagenesis. Oligonucleotides used in this study are listed in Supplementary Table 1.

The antiserum was generated by immunizing rabbits with a polypeptide comprising the C-terminal 20 residues of ORF2p, and the antibody was purified by using an ORF2p antigen column (HuaBio, Hangzhou, China). An anti-Enterovirus 71 VP1 antibody (dilution 1:1000; GTX132338) and anti-Enterovirus D68 VP1 antibody (dilution 1:1000; GTX132313) were purchased from GeneTex (San Antonio, USA). The monoclonal mouse anti-α-tubulin antibody (dilution 1:2000; A01410) was purchased from GenScript (Piscataway, USA). A polyclonal rabbit anti-HA antibody (dilution 1:2000; 71-5500) was purchased from Thermo Fisher Scientific (Rochester, NY, USA). HA-Tag (6E2) Mouse mAb (dilution 1:800; Alexa Fluor® 488 Conjugate) (2350) and ProLong® Gold Antifade Reagent with DAPI (8961) were from Cell Signalling Technology, Inc. (Minnesota, USA). An anti-LC3B antibody (dilution 1:1000; L7543) was purchased from Sigma (Darmstadt, Germany).

**Cells**. HT-29 human IECs (Cell Bank of the Chinese Academy of Sciences, TCHu103) were cultured in McCoy's 5a Medium supplemented with 10% foetal bovine serum (FBS) and penicillin/streptomycin solution. LS 174 T human IECs (ATCC, CL-188), 293 T human embryonic kidney cells (ATCC, CRL-3216), RD human rhabdomyosarcoma cells (ATCC, CCL-136), HeLa human cervical

epithelial cells (ATCC, CCL-2), Vero African green monkey kidney cells (ATCC, CCL-81), HepG2 human hepatocellular carcinoma cells (ATCC, HB-8065), A549 (ATCC, CRM-CCL-185), and NSC-34 mouse motor neuron cells (Cedarlane Laboratories, CLU140) were cultured in Dulbecco's Modified Eagle's Medium supplemented with 10% FBS and penicillin/streptomycin solution. U937 human lymphoma cells (ATCC, CRL-1593.2), HCT-8 human IECs (ATCC, CCL-244), Hce-8693 human IECs (Cell Bank of the Chinese Academy of Sciences, TCHu 70) and LS513 human IECs (Cell Bank of the Chinese Academy of Sciences, TCHu237) were cultured in RPMI-1640 medium supplemented with 10% FBS and penicillin/streptomycin solution. U937 cells were differentiated by addition of 100 nM phorbol-12-myristate-13-acetate (PMA) for 48 h. LoVo human IECs (Cell Bank of the Chinese Academy of Sciences, TCHu 82) were cultured in F-12K Medium supplemented with 10% foetal bovine serum (FBS) and penicillin/streptomycin solution. RKO human IECs (ATCC, CRL-2577) were cultured in Eagle's Minimum Essential Medium supplemented with 10% foetal bovine serum (FBS) and penicillin/streptomycin solution.

Human IECs were provided by Fenghbio, Inc. (Changsha, China). All studies were approved by the Ethics Committee of the Institute of Virology and AIDS Research, First Hospital of Jilin University. Written informed consent was obtained from the parents involved in our study. For human IEC isolation, fresh normal human tissue surrounding intestinal carcinoma was obtained from a cancer donor and washed three times with wash buffer (1% penicillin/streptomycin in PBS buffer). The human tissue was dissected into 1 mm segments and then transferred to collagenase digestion buffer for 30 min at 37 °C. Cells were centrifuged at 1000 g for 5 min, and the pelleted cells were washed, resuspended in fresh culture medium, and cultured at 37 °C with 5% $CO_2$.

To generate ORF2p-expressing HT-29 cells, we co-transfected HEK 293 T cells with pCDH, pCDH-ORF2p plus pRSV-Rev, pMDLg/pRRE, and pCMV-VSV-G using Lipofectamine 2000 (Invitrogen) according to the manufacturer's instructions. Two days after transfection, the cell culture medium was harvested. Cell debris was removed by centrifugation at $10,000 \times g$ for 5 min, and supernatants were stored at −80 °C. Supernatants were incubated with HT-29 cells for 6 h, and the medium was then replaced with fresh culture medium. Transduced cells were selected in 10% FBS-DMEM supplemented with puromycin (2 μg/ml; Sigma) one day after infection.

**Viruses**. EV-D68 prototype Fermon (ATCC, VR-1826) was propagated in RD cells. EV-D68 viruses in the supernatants of infected cells were harvested, clarified by low-speed centrifugation and passed through a 0.22-mm filter, and viral particles were pelleted through a 20% sucrose cushion in an SW28 rotor at 28,000 rpm for 90 min. Purified virions were stored at −80 °C.

**Virus recovery from infectious cDNA**. *In vitro* synthesized RNA transcripts were obtained using a RiboMAX™ Large Scale RNA Production Systems-Sp6 kit (Madison, Promega) with MluI-linearized pA12-EV-A71 or a mutant clone as a template. The resultant RNAs were transfected into RD cells with Lipofectamine 3000 according to the manufacturer's instructions.

**Viral titre assay**. Viral titres were determined by the appearance of CPE in RD cells using microtitration analysis according to the Reed-Muench method[46]. Viral titres were expressed as the 50% tissue culture infectious dose (TCID50).

**Immunoblotting**. Cell samples were harvested by scraping, washed twice with cold PBS, lysed in lysis buffer (150 mM Tris, pH 7.5, with 150 mM NaCl, 1% Triton X-100, and complete protease inhibitor cocktail tablets [Roche]) at 4 °C for 30 min, and centrifuged at 10,000 g for 30 min. The supernatants were mixed with 1X loading buffer (0.08 M Tris, pH 6.8, with 2.0% SDS, 10% glycerol, 0.1 M DTT, and 0.2% bromophenol blue) and boiled for 5 min. The cell lysates were separated via SDS-PAGE and transferred to nitrocellulose membranes using a semidry apparatus (Bio-Rad). The membranes were probed with various primary antibodies against the proteins of interest; secondary antibodies were alkaline phosphatase-conjugated anti-goat IgG and anti-mouse IgG (Jackson ImmunoResearch Laboratories). Staining was conducted with 5-bromo-4-chloro-3-indolyl phosphate and NBT solutions prepared from chemicals obtained from Sigma-Aldrich (Milwaukee, USA). The uncropped blots are provided as a Source Data file.

**Quantitative real-time PCR (qRT-PCR)**. Total RNA from cells was isolated using TRIzol (Life Technologies) according to the manufacturer's instructions, including the DNase I digestion step. Samples were incubated in 10 μl diethyl pyrocarbonate (DEPC)-treated water with 1x RQ1 RNase-Free DNase buffer, l μl RQ1 RNase-free DNase (Promega), and 4 U RNase inhibitor (New England Biolabs) for 30 min at 37 °C. DNase activity was inactivated by the addition of 1 μl RQ1 DNase stop solution and incubation at 65 °C for 10 min. RNA was reverse transcribed by using random primers and Multiscribe reverse transcriptase from a High-Capacity cDNA Archive kit (Applied Biosystems) according to the manufacturer's instructions. The cDNA was either used undiluted or serially diluted in DEPC-treated water before real-time PCR to ensure that the amplification was within the linear range of detection. A StepOne Real-Time PCR System (Applied Biosystems, Carlsbad, CA) was used for qRT-PCR amplifications. The reactions were performed under the

following conditions: 50 °C for 2 min and 95 °C for 10 min; 40 cycles of 95 °C for 15 s and 60 °C for 1 min; and a dissociation protocol. Single peaks in the melting curve analysis indicated specific amplicons.

**Luciferase assay.** To evaluate the viral transcriptional activity of EV-A71 and EV-A71ΔORF2p, we constructed a promoter-driven firefly luciferase plasmid p5′UTR WT-pGL3, p5′UTR ΔORF2p-pGL3 and expression plasmid p3D-VR1012. p5′UTR-Luc (200 ng) was transfected or co-transfected with p3D-VR1012 (800 ng) into cells[47]. Luciferase activity was assessed 48 h after transfection. Briefly, the cells were harvested and lysed with passive lysis buffer and then centrifuged at $12000 \times g$ for 10 min. Supernatants and luciferase substrate (Promega, E190) were mixed in a 96-well plate, and fluorescence was quantified with a Fluoroskan Ascent^TM FL instrument (Thermo Fisher, 5210450).

**Immunostaining and confocal microscopy.** HT-29 cells were transfected with pEV-A71 ORF2p-HA. After 36 h, treated cells were transferred to coverslips overnight and subsequently fixed for 15 min with 4% paraformaldehyde in PBS, permeabilized for 10 min in 0.1% Triton X-100 in PBS, and blocked using 5% BSA for 1 h. Then, the cells were incubated with HA-Tag (6E2) Mouse mAb (Alexa Fluor® 488 Conjugate) at 4 °C for 16 h. Nuclei were counterstained with 4,6-diamidino-2-phenylindole (DAPI). Images were captured using a ZEISS laser scanning confocal microscope. ZEISS ZEN Microscope software was used for acquisition. For live cell imaging, HT-29 cells were transfected in coverslip glass-bottomed cell culture dishes and then visualized after 24 h under an Olympus FV3000 confocal laser scanning microscope (Olympus; Tokyo, Japan). All images were acquired using a ×63 objective, and image analysis and manipulation were performed using ImageJ software.

**Viral attachment assays.** For viral attachment experiments, cells were first washed with cold DMEM, and then EV-A71 viruses were added to the cells. After incubation at 4 °C or 37 °C for 2 h, treated cells were washed with cold DMEM to remove unbound viruses. Total RNA was extracted using an RNeasy Mini Kit (Qiagen). The bound virus RNA was determined by using qRT-PCR.

**Intraendosomal pH determination.** We used amine-reactive pHrodo dyes (Life Technologies, cat no. P35368) to detect changes in the pH of virus-containing endosomes[47]. EV-A71 virus was purified and dissolved in PBS and incubated with amine-reactive pHrodo dyes for 40 min at room temperature in the dark, then repurified by pelleting through a sucrose cushion before infection. Next, cells were seeded on glass plates overnight. Then, the cells were infected with dye-conjugated EV-A71 virus at 4 °C for 30 min, washed with PBS, incubated at 37 °C, and observed at the indicated time point with a confocal microscope.

**Neonatal mouse infection model.** Specific-pathogen-free (SPF) ICR neonatal mice within 24 h of birth (Experimental Animal Center, College of Basic Medicine, Jilin University) were used to establish the animal model of viral infection. All animal protocols were approved by the institutional Animal Care and Use Committee and were strictly followed. All efforts were made to minimize animal suffering. Neonatal mice were randomly divided into 5 groups ($n = 8$–10 per group) and inoculated intracerebrally with two different doses of the EV-A71 virus, EV-A71ΔORF2p virus or DMEM. Survival rates were monitored daily for 20 days post-infection. Control mice remained healthy throughout the experiments. All animal experiments were conducted according to animal protocols approved by the Insititute of Virology and AIDS Research Subcommittee of Research Animal Care.

**Statistical analysis.** Differences among test groups were analysed by ANOVA (Stata Corp, College Station, TX). A value of $p < 0.05$ was considered significant.

**Reporting Summary.** Further information on research design is available in the Nature Research Reporting Summary linked to this article.

## Data availability
All relevant data are available from the corresponding author upon reasonable request. The source data underlying Figs. 1c, 2a–e, 3a, c, d, 4a–f and h, i and Supplementary Figs. 1c–f, 2c, 3, 5, 6a, b, d, e, 7, 8b, 9b–e and 10 are provided as a Source Data file.

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

## Acknowledgements

We thank H. Xu, J. Yang, D. Wang, Y. Li, and G. Zhou for technical assistance, Drs. T. Cheng, and XF. Yu for key reagents. This work was supported in part by funding from the National Natural Science Foundation of China (81772183, 31600132 and 31800150), the Department of Science and Technology of Jilin Province (No. 20190304033YY and 20180101127JC), the open project of Key Laboratory of Organ Regeneration and Transplantation, Ministry of Education, the Program for JLU Science and Technology Innovative Research Team (2017TD-08) and Fundamental Research Funds for the Central Universities.

## Author contributions

H.G., Y.L., G.L., Y.J., Y.S., and R.B. performed the experiments. W.W., H.G., H.H., T.C., and C.W. analysed the data. W.W. wrote the paper with help from all authors. W.W. directed the project.

## Additional information

**Competing interests:** The authors declare no competing interests.

