## [Peer Review File · Nature Communications]

Editorial Note: This manuscript has been previously reviewed at another journal that is not operating a transparent peer review scheme. This document only contains reviewer comments and rebuttal letters for versions considered at Nature Communications .

Reviewers' Comments:

Reviewer #1:

Remarks to the Author:

Nature Comm 2-2019

Guo... Wei

A second open reading frame in human enterovirus determines viral replication in intestinal epithelial cells

I reviewed a previous version of this paper. This revised paper includes new data, in particular the demonstration that orf2 protein is needed for release of mature virions from primary intestinal epithelial cells, and that orf2p appears to associate with autophagosomal vesicles and be may be involved somehow in the regulation of autophagy in infected cells. As I said before, the paper is novel and exciting.

I still think some experimental data are missing, and that revision is needed to correct confusion and to improve readability. In addition to the points discussed below, I have made a number of suggested changes to the manuscript itself. I don't want to be writing the authors' paper, but in many cases it was easier to just make some changes directly.

Substantive comments:

1. Although the IEC experiment is overall convincing, the levels of intracellular RNA need to be shown in 4a.
2. Lines 170-184 are very confusing. If you want to make the point that there's a specific block to particle release in HT-29 cells and that it's relieved by ORF2p, measure RNA and protein synthesis, and release of infectious viral particles after freeze thaw. Don't jump back and forth from one figure to another, and don't leave out the virus titers in cells and supernatant. I do not find Extended Figure 7a convincing as evidence that there's a block to virus release. The difference in RNA levels is less than 10-fold.
3. I still don't think the panel of cell lines examined is quite sufficient to say there's a specific effect in intestinal epithelial cells. The titer differences in LS174T intestinal cells is small, and a bigger effect is seen in NSC34 cells, a mouse neuron. It would help to look at additional human intestinal cells, but it might be better to soften the writing. The primary IECs ARE undoubtedly a good model of human intestinal cells.
4. The mouse data should be in the results, not just the discussion; you might want to save them for a separate paper, as they do not really support the point you are making here; similarly, the infection of the NSC-34 cells is not consistent with the point you want to make.
5. The entry experiment described in Fig 6d is not fully described and needs additional controls. Was the virus repurified after labeling with dye, or was the virus added to cells in the presence of free dye. If the latter, you need a control to show that the free dye is not itself internalized and trafficked to endosomes; it could be labeling surface proteins that are then endocytosed. Plus, there should be a

control with cells incubated in the cold, to show that non-internalized virus doesn't light up.

6. The role of orf2 in stimulating autophagy is important, and the data in Ext Fig 9a should be quantified (how many puncta per cell in ORF2+ and ORF2- cells). In 9b, this really isn't quantitation. It's just a demonstration of colocalization.

7. The discussion is rambling and disappointing. It really needs revision to bring out the important points. The really exciting things here are the identification of a new orf, the role in virus exit, the possible relation to autophagy, and the potential importance for replication in epithelial cells in the intestine. Interesting secondary things are the known role of the 5-UTR in tropism for other cell types, and speculation about a possible role for orf2 in virulence and tropism for other tissues (this could tie in the mouse data if you choose to show them).

Minor additional points:

In general, experimental results should be in past tense, conclusions and known facts in present tense.

Be sure to put figure citations in the text where they are mentioned, not in the next sentence.

Lines 134-140. I don't know that anything here is "crucial," although I do think it does support the argument for a role in intestinal replication. The isolation of EV68 from stool, and a possible role for mixed infection seems somewhat irrelevant here, and might be rephrased and moved to discussion.

Line 148: WIG/AAA and HPV/AA are a bit confusing. I assume the first is AAAHPV and the second is WIGAAA. Please make clearer

Fig 2. Legend, define HT-29-CDH

Fig 4b. I assume the differences in frozen/thawed are not significant. ? Specify?

Ext Fig 1d. In legend, specify top panel is EV68 and bottom is EV71.

Ext Fig6c. From legend it's hard to tell what was done here. Clearer if you say virus was "labeled" with the dye (and see comments above).

Jeffrey Bergelson

Reviewer #3:

Remarks to the Author:

In this manuscript, Guo and colleagues present data indicating that there is a second ORF in EV71 that is important for viral replication in intestinal epithelial cells. The manuscript builds on some very interesting observations, but overall the data contain several flaws.

Most importantly, as also raised by reviewer #2 as specific comment no 1, the outcomes of nearly all experiments are difficult to interpret because they were done at low MOI infection and analysed at 48

to 72 h after infection, implicating that the infection involved multiple rounds of infection. In response to this comment, the authors have performed only one single-cycle experiment, which is shown in Extended Data Figure 7. Here they only analysed virus titers in the supernatants of HT-29 cells. Why did they not include the cellular titers, to show that similar amounts of virus were produced. In addition, why did they not include RD cells which are important controls

However, the issues about the low versus high MOI infections remain. For example in Fig 4B, where the authors show virus titers obtained in a transfection experiment. How is it possible that upon RNA transfection the amount of ORF2p-lacking viral RNA in cells is higher than of wt virus at 48 h posttransfection? The ORF2p virus should have a defect in release and spreading and should therefore be less capable in spreading to other cells, and should therefore have lower viral RNA levels. Here, as with other low MOI infection studies, the analysis is blurred because the experiment involves many rounds of replication. This example shows how important it is to analyse virus titers after a single round of replication (i.e. high MOI experiments).

The authors failed to adequately respond to comment no 2, where reviewer #2 asks for additional support for a role of ORF2p in particle release. Instead of providing this information, the authors only exclude effects on a number of other steps in the viral life cycle.

The authors also failed to adequately respond to comment no 3, where reviewer #2 asks for more detailed experiment to quantify effects on translation and replication. The authors do not show any experiments to study replication, e.g. with replicons, which is definitely needed and thereby a serious omission.

With regard to comment no 5, the evidence that is provided suggesting that ORF2p localizes to the extracellular vesicles and that it regulates transport and/or release of these vesicles is far from convincing. At best, the authors show some preliminary data that point to such functions (e.g. colocalization with LC3 and the wortmannin experiment, although this latter is not convincing), but experimental evidence that provides firm support for their suggestions is lacking.

Overall, this manuscript is lacking the in depth and mechanistic quality that is provided by Nature journals. The manuscript is still in a preliminary stage. There is some evidence that ORF2p is important for virus spreading, but the attempts undertaken to obtain more in depth mechanistic insights remain superficial.

Reviewers' comments:

Reviewer #1 (Remarks to the Author):

Nature Comm 2-2019

Guo... Wei

A second open reading frame in human enterovirus determines viral replication in intestinal epithelial cells

I reviewed a previous version of this paper. This revised paper includes new data, in particular the demonstration that orf2 protein is needed for release of mature virions from primary intestinal epithelial cells, and that orf2p appears to associate with autophagosomal vesicles and be may be involved somehow in the regulation of autophagy in infected cells. As I said before, the paper is novel and exciting.

I still think some experimental data are missing, and that revision is needed to correct confusion and to improve readability. In addition to the points discussed below, I have made a number of suggested changes to the manuscript itself. I don't want to be writing the authors' paper, but in many cases it was easier to just make some changes directly.

Response: We greatly appreciate the recognition of the potential importance of our discoveries by the reviewer and find the comments both insightful and helpful. Below is our point-by-point response to the reviewer's critiques.

Substantive comments:

1. Although the IEC experiment is overall convincing, the levels of intracellular RNA need to be shown in 4a.

Response: As suggested, we have added these data to the revised manuscript as a new Figure 4b.

2. Lines 170-184 are very confusing. If you want to make the point that there's a specific block to particle release in HT-29 cells and that it's relieved by ORF2p, measure RNA and protein synthesis, and release of infectious viral particles after freeze thaw. Don't jump back and forth from one figure to another, and don't leave out the virus titers in cells and supernatant. I do not find Extended Figure 7a convincing as evidence that there's a block to virus release. The difference in RNA levels is less than 10-fold.

Response: We thank the reviewer for indicating this issue. We have corrected the description in the manuscript (Lines 170-176).

3. I still don't think the panel of cell lines examined is quite sufficient to say there's a specific effect in intestinal epithelial cells. The titer differences in LS174T intestinal cells is small, and a bigger effect is seen in NSC34 cells, a mouse neuron. It would help to look at additional human intestinal cells, but it might be better to soften the writing. The primary IECs ARE undoubtedly a good model of human intestinal cells.

Response: As suggested, we have investigated the roles of ORF2p in virus infection in different intestinal epithelial cells (LS 174T, LoVo, RKO, HCT-8, LS 513, Hce-8693 cells) by EV-A71 at an MOI of 10. We have incorporated these data in Supplementary Figure 3 and corrected our description in the manuscript (Lines 116-118).

4. The mouse data should be in the results, not just the discussion; you might want to save them for a separate paper, as they do not really support the point you are making here; similarly, the infection of the NSC-34 cells is not consistent with the point you want to make.

Response: As suggested by the reviewer, we have incorporated the mouse data into the results (Lines 207-210) and removed the data involving NSC-34 cells.

5. The entry experiment described in Fig 6d is not fully described and needs additional controls. Was the virus repurified after labeling with dye, or was the virus added to cells in the presence of free dye. If the latter, you need a control to show that the free dye is not itself internalized and trafficked to endosomes; it could be labeling surface proteins that are then endocytosed. Plus, there should be a control with cells incubated in the cold, to show that non-internalized virus doesn't light up.

Response: The EV-A71 and EV-A71 Δ ORF2p viruses used in the experiment were repurified after incubation with amine-reactive pHrodo dyes; we did not detect an obvious fluorescent signal in the control samples. As suggested, we have corrected the methods in the manuscript and incorporated the control into revised Supplementary Figure 6c.

6. The role of orf2 in stimulating autophagy is important, and the data in Ext Fig 9a should be quantified (how many puncta per cell in ORF2+ and ORF2- cells). In 9b, this really isn't quantitation. It's just a demonstration of colocalization.

Response: We thank the reviewer for this suggestion. The required data have been added to the revised figures (Supplementary Figure 9b).

7. The discussion is rambling and disappointing. It really needs revision to bring out the important points. The really exciting things here are the identification of a new orf, the role in virus exit, the possible relation to autophagy, and the potential importance for replication in epithelial cells in the intestine. Interesting secondary things are the known role of the 5-UTR in tropism for other cell types, and speculation about a possible role for orf2 in virulence and tropism for other tissues (this could tie in the mouse data if you choose to show them).

Response: We greatly appreciate the constructive criticism from the reviewer. We have revised the discussion as recommended (Lines 213-263).

Minor additional points:

In general, experimental results should be in past tense, conclusions and known facts in present tense.

Response: As suggested by the reviewer, we have checked and corrected the verb tense in the manuscript.

Be sure to put figure citations in the text where they are mentioned, not in the next sentence.

Response: We thank the reviewer for the suggestion; we have made these corrections.

Lines 134-140. I don't know that anything here is "crucial," although I do think it does support the argument for a role in intestinal replication. The isolation of EV68 from stool, and a possible role for mixed infection seems somewhat irrelevant here, and might be rephrased and moved to discussion.

Response: We have changed our description as suggested (Lines 138-140).

Line 148: WIG/AAA and HPV/AA are a bit confusing. I assume the first is AAAHPV and the second is WIGAAA. Please make clearer

Response: As suggested by the reviewer, we have incorporated the description of these two mutants into the manuscript (Lines 146-149).

Fig 2. Legend, define HT-29-CDH

Response: The control HT-29-CDH cells were generated by transducing them with the empty lentiviral vector pCDH-CMV-MCS-EF1-Puro. As suggested by the reviewer, we have incorporated this description into the manuscript (Lines 417-418).

Fig 4b. I assume the differences in frozen/thawed are not significant. ? Specify?

Response: The reviewer is correct, and we have labelled this in revised Figure 4c.

Ext Fig 1d. In legend, specify top panel is EV68 and bottom is EV71.

Response: We have corrected these items in the manuscript (Lines 467-468).

Ext Fig6c. From legend it's hard to tell what was done here. Clearer if you say virus was "labeled" with the dye (and see comments above).

Response: We thank the reviewer for this suggestion. To address this issue, we have corrected the figure legends and incorporated a schematic diagram in revised Supplementary Figure 6c.

Jeffrey Bergelson

Response: Thank you very much. We are deeply grateful for the high-quality review.

Reviewer #3 (Remarks to the Author):

In this manuscript, Guo and colleagues present data indicating that there is a second ORF in EV71 that is important for viral replication in intestinal epithelial cells. The manuscript builds on some very interesting observations, but overall the data contain several flaws.

Response: We thank the reviewer for the careful reading and comments. We have taken these points into serious consideration and revised the manuscript accordingly. Specific comments are addressed below.

Most importantly, as also raised by reviewer #2 as specific comment no 1, the outcomes of nearly all experiments are difficult to interpret because they were done at low MOI infection and analysed at 48 to 72 h after infection, implicating that the infection involved

multiple rounds of infection. In response to this comment, the authors have performed only one single-cycle experiment, which is shown in Extended Data Figure 7. Here they only analysed virus titers in the supernatants of HT-29 cells. Why did they not include the cellular titers, to show that similar amounts of virus were produced. In addition, why did they not include RD cells which are important controls

Response: As suggested by the reviewer, we have incorporated all the recommended data into revised Figure 4d and Supplementary Figure 7.

However, the issues about the low versus high MOI infections remain. For example in Fig 4B, where the authors show virus titers obtained in a transfection experiment. How is it possible that upon RNA transfection the amount of ORF2p-lacking viral RNA in cells is higher than of wt virus at 48 h posttransfection? The ORF2p virus should have a defect in release and spreading and should therefore be less capable in spreading to other cells, and should therefore have lower viral RNA levels. Here, as with other low MOI infection studies, the analysis is blurred because the experiment involves many rounds of replication. This example shows how important it is to analyse virus titers after a single round of replication (i.e. high MOI experiments).

Response: We thank the reviewer for pointing out this issue. The suggestion that we assess the virus titre after a single round of replication has greatly strengthened our conclusions. To address the issue raised, we have performed the recommended one-step growth curve experiments for EV-A71 and EV-A71 Δ ORF2p viruses in RD, HT-29, and primary IECs at an MOI of 10 (new Figure 2a, 2b, 2c). The results indicate that ORF2p expression is required for the full replication capacity of EV-A71. In addition, we further confirmed that ectopic expression of ORF2p could restore replication of the ORF2p-defective EV-A71 Δ ORF2p viruses in HT-29 cells at a high MOI (new Figure 2d). Lacking ORF2p expression significantly decreased EV-A71 Δ ORF2p (MOI=10) replication in diverse intestinal cell lines compared to that of EV-A71 wild-type viruses (MOI=10) (Supplementary Figure 3).

The authors failed to adequately respond to comment no 2, where reviewer #2 asks for additional support for a role of ORF2p in particle release. Instead of providing this information, the authors only exclude effects on a number of other steps in the viral life cycle.

Response: We apologize for our poor description. We have performed the experiments and confirmed that ORF2p is required for viral RNA and infectious particle release from EV-A17 RNA transfected IECs (Figure 4a-c). ORF2p deficiency specifically decreased the amount of infectious EV-A71 Δ ORF2p virus particles in the supernatant (Figure 4d,4e). Moreover, ectopic expression of ORF2p restored accumulation of the viral structural protein VP1 in the supernatant of EV-A71 Δ ORF2p-infected cells (Figure 4f). We have incorporated the required data into the manuscript (Lines 158-179).

The authors also failed to adequately respond to comment no 3, where reviewer #2 asks for more detailed experiment to quantify effects on translation and replication. The authors do not show any experiments to study replication, e.g. with replicons, which is definitely needed and thereby a serious omission.

Response: We greatly appreciate the reviewer for this suggestion. We have generated luciferase replicons for EV-A71 and EV-A71 Δ ORF2p by replacing the P1 structure

region with the firefly luciferase gene. The EV-A71 and EV-A71 Δ ORF2p pseudoviruses were packaged in HEK293T cells. RD and HT-29 cells were incubated with equal amounts of pseudoviruses and harvested at 9 h post-infection for luciferase activity assays. The results demonstrate that a lack of ORF2p did not significantly influence the early steps of enterovirus replication (Supplementary Figure 6e).

With regard to comment no 5, the evidence that is provided suggesting that ORF2p localizes to the extracellular vesicles and that it regulates transport and/or release of these vesicles is far from convincing. At best, the authors show some preliminary data that point to such functions (e.g. colocalization with LC3 and the wortmannin experiment, although this latter is not convincing), but experimental evidence that provides firm support for their suggestions is lacking.

Response: We thank the reviewer for this comment. The product of the newly identified ORF of human enteroviruses in the present study, i.e., ORF2p, contributes to full replication capacity in the intestinal cells by facilitating virus particle release.

Considering the importance of cell autophagy in enterovirus replication, the identification that ORF2p stimulates autophagy in EV-A71-infected or ORF2p-overexpressing cells should be helpful for a better understanding of the pathogenesis of enteroviruses. The mechanisms of ORF2p-triggered autophagy and its roles in virus replication still need to be further investigated in future studies. We discuss the reviewer's comment in the revised manuscript (Page 7, lines 230-242).

Overall, this manuscript is lacking the in depth and mechanistic quality that is provided by Nature journals. The manuscript is still in a preliminary stage. There is some evidence that ORF2p is important for virus spreading, but the attempts undertaken to obtain more in depth mechanistic insights remain superficial.

Response: We thank the reviewer for these comments. We have addressed the issues raised by the reviewers, which has greatly improved our manuscript.

Reviewers' Comments:

Reviewer #1:

Remarks to the Author:

In this revision, the authors have addressed all of my substantive concerns. I think the paper is convincing and important to the field.

I have only a few minor comments, mostly about the writing.

1. In Supplemental Figure 3, deletion of Orf2p has an effect on replication in all the cell lines, but it is much greater in some than in others. I would suggest mentioning this in the text (lines 117-118). Something like "when we examined EV71 replication in a panel of intestinal epithelial cell lines, we found that deletion of orf2p caused a marked reduction of virus titers in several cell lines, and smaller, but still significant reductions in the others".

2. In the experiment with labeled virus, the text still does not specifically say the virus was repurified after labeling. Lines 721-25: "EVA71 was incubated withdye overnight, then repurified by pelleting through a sucrose cushion" or something like that.

3. Line 544-45 (quantification of LC3 puncta). Specify number of cells examined.

4. Lines 165, 172: equal amountS

5. Line 174-175. Consistent with this.. (consistently means something else); In a control experiment, there was...

6. Line 224: demonstrating that orf2p HAS

7. Line 241: why "urgently"?

8. Line 247. which LEADS? us to speculate that

Jeffrey Bergelson

Reviewer #3:

Remarks to the Author:

Comment: Most importantly, as also raised by reviewer #2 as specific comment no 1, the outcomes of nearly all experiments are difficult to interpret because they were done at low MOI infection and analysed at 48 to 72 h after infection, implicating that the infection involved 4 multiple rounds of infection. In response to this comment, the authors have performed only one single-cycle experiment, which is shown in Extended Data Figure 7. Here they only analysed virus titers in the supernatants of HT-29 cells. Why did they not include the cellular titers, to show that similar amounts of virus were produced. In addition, why did they not include RD cells which are important controls

Response: As suggested by the reviewer, we have incorporated all the recommended data into revised Figure 4d and Supplementary Figure 7.

Reviewer #3: First, the authors have NOT incorporated ALL the recommended data in Figure 4d. Initially all experiments were performed at low MOI and then the readout was done at 48 post

infection which involves multiple rounds of replication. This was a serious omission that was noted by reviewer #2. In response to that question, the authors only come up with one single experiment that was shown in Suppl Fig 7 (HT-29 cells), which was disappointing as it would have been important to have performed single and multicycle experiments throughout the manuscript in all Figures. In response to my question on virus titers, the authors have now completely deleted Supp Fig S7, and come up with revised Fig 4D, which is on IEC cells, and a new Suppl Fig S7, which is on RD cells. They don't show anything for HT-29 cells for reasons that are not explained.....

Comment: However, the issues about the low versus high MOI infections remain. For example in Fig 4B, where the authors show virus titers obtained in a transfection experiment. How is it possible that upon RNA transfection the amount of ORF2p-lacking viral RNA in cells is higher than of wt virus at 48 h posttransfection? The ORF2p virus should have a defect in release and spreading and should therefore be less capable in spreading to other cells and should therefore have lower viral RNA levels. Here, as with other low MOI infection studies, the analysis is blurred because the experiment involves many rounds of replication. This example shows how important it is to analyse virus titers after a single round of replication (i.e. high MOI experiments).

Response: We thank the reviewer for pointing out this issue. The suggestion that we assess the virus titre after a single round of replication has greatly strengthened our conclusions. To address the issue raised, we have performed the recommended onestep growth curve experiments for EV-A71 and EV-A71 Δ ORF2p viruses in RD, HT-29, and primary IECs at an MOI of 10 (new Figure 2a, 2b, 2c). The results indicate that ORF2p expression is required for the full replication capacity of EV-A71. In addition, we further confirmed that ectopic expression of ORF2p could restore replication of the ORF2p-defective EV-A71 Δ ORF2p viruses in HT-29 cells at a high MOI (new Figure 2d). Lacking ORF2p expression significantly decreased EV-A71 Δ ORF2p (MOI=10) replication in diverse intestinal cell lines compared to that of EV-A71 wild-type viruses (MOI=10) (Supplementary Figure 3).

Reviewer #3: It is kind that the authors thank me for pointing out this issue, but I cannot understand at all that they did not see this themselves and makes me wonder whether they have full understanding of the design and interpretation of their experiments. It remains a mystery why there is no significant difference in the IEC cells (i.e. the frozen-thawed lysates) infected with wt and ORF2p lacking virus at 48 h posttransfection. In this RNA transfection experiment, virus production also likely involves multiple rounds of replication. As ORF2p-lacking virus release is severely impaired, this should have resulted in reduced levels in the ORF2p-lacking virus infected cell culture. Because no virus titers have been determined at various times posttransfection between 12 and 48 h, this remains a very poor experiment.

Similar as my comment on Fig. 4B, also data shown in Figs 4c-e of the previous manuscript suffered from wrong set up. In Fig 4C, the authors found higher virus titers in cells infected with ORF2p lacking virus, which does not make any sense when the virus has a defect in release. Where have these figures gone? They have not been corrected. Instead, the authors have just removed them from the new version without any explanation. It seems that authors prefer to selectively include results that match their ideas, while leaving out results that do not match with their message...

The authors have included new Figs 2a-c in their new manuscript. The results shown in 2c (growth in IECs) differ significantly from those in the previous Fig 2c that was included in the previous manuscript. Previously, they found that replication was similar for wt and ORF2p-lacking virus up to 12 h, thereafter there was a difference (which was in line with no role in RNA replication but in virus release). Instead now, they show that virus replication of the ORF2p virus is severely impaired in the

ORF2p-lacking virus. This demonstrates very poor reproducibility of the results provided by these authors. Furthermore, the data in the new Fig 2c (where there is a 500-fold difference in virus titers between wt and ORF2p-lacking virus) are inconsistent with those in new Fig 4d, where it is shown that they replicate to similar titers. Overall, the data seem poorly reproducible and inconsistent.

To make their point that ORF2p is important for virus release in intestinal cells, the authors have incorporated more intestinal cell lines in Suppl Fig 3. However, they have replaced the old Suppl Fig 3, which showed that ORF2p is also important for release in NSC-34 cells, which are mouse neural cells (as remarked by Reviewer #1). So, it seems that the authors without any scruples just remove data that do not fit their hypothesis, rather than discussing and softening their conclusions. This is a big disgrace and can mislead the readership!

Comment: The authors failed to adequately respond to comment no 2, where reviewer #2 asks for additional support for a role of ORF2p in particle release. Instead of providing this information, the authors only exclude effects on a number of other steps in the viral life cycle.

Response: We apologize for our poor description. We have performed the experiments and confirmed that ORF2p is required for viral RNA and infectious particle release from EV-A17 RNA transfected IECs (Figure 4a-c). ORF2p deficiency specifically decreased the amount of infectious EV-A71 Δ ORF2p virus particles in the supernatant (Figure 4d,4e). Moreover, ectopic expression of ORF2p restored accumulation of the viral structural protein VP1 in the supernatant of EV-A71 Δ ORF2p-infected cells (Figure 4f). We have incorporated the required data into the manuscript (Lines 158-179).

Reviewer #3: Okay

Comment: The authors also failed to adequately respond to comment no 3, where reviewer #2 asks for more detailed experiment to quantify effects on translation and replication. The authors do not show any experiments to study replication, e.g. with replicons, which is definitely needed and thereby a serious omission.

Response: We greatly appreciate the reviewer for this suggestion. We have generated luciferase replicons for EV-A71 and EV-A71 Δ ORF2p by replacing the P1 structure 5 region with the firefly luciferase gene. The EV-A71 and EV-A71 Δ ORF2p pseudoviruses were packaged in HEK293T cells. RD and HT-29 cells were incubated with equal amounts of pseudoviruses and harvested at 9 h post-infection for luciferase activity assays. The results demonstrate that a lack of ORF2p did not significantly influence the early steps of enterovirus replication (Supplementary Figure 6e).

Reviewer #3: Okay

Comment: With regard to comment no 5, the evidence that is provided suggesting that ORF2p localizes to the extracellular vesicles and that it regulates transport and/or release of these vesicles is far from convincing. At best, the authors show some preliminary data that point to such functions (e.g. colocalization with LC3 and the wortmannin experiment, although this latter is not convincing), but experimental evidence that provides firm support for their suggestions is lacking.

Response: We thank the reviewer for this comment. The product of the newly identified ORF of human

enteroviruses in the present study, i.e., ORF2p, contributes to full replication capacity in the intestinal cells by facilitating virus particle release. Considering the importance of cell autophagy in enterovirus replication, the identification that ORF2p stimulates autophagy in EV-A71-infected or ORF2p-overexpressing cells should be helpful for a better understanding of the pathogenesis of enteroviruses. The mechanisms of ORF2p-triggered autophagy and its roles in virus replication still need to be further investigated in future studies. We discuss the reviewer's comment in the revised manuscript (Page 7, lines 230-242).

Reviewer #3: The authors spend a few lines of text, but still I am not convinced by their data. The data remain superficial. Unfortunately, no new attempts to provide stronger data and to convince the readership have been undertaken.

Reviewers' comments:

Reviewer #1 (Remarks to the Author):

In this revision, the authors have addressed all of my substantive concerns. I think the paper is convincing and important to the field.

Response: We greatly appreciate the recognition of the potential importance of our discoveries by the reviewer.

I have only a few minor comments, mostly about the writing.

1. In Supplemental Figure 3, deletion of Orf2p has an effect on replication in all the cell lines, but it is much greater in some than in others. I would suggest mentioning this in the text (lines 117-118). Something like "when we examined EV71 replication in a panel of intestinal epithelial cell lines, we found that deletion of orf2p caused a marked reduction of virus titers in several cell lines, and smaller, but still significant reductions in the others".

Response: We appreciate the suggestion by the reviewer, and we have incorporated a description of this observation in the revised manuscript (page 4, lines 117-119).

2. In the experiment with labeled virus, the text still does not specifically say the virus was repurified after labeling. Lines 721-25: "EVA71 was incubated withdye overnight, then repurified by pelleting through a sucrose cushion" or something like that.

Response: We have incorporated the information in the revised manuscript (page 29, line 734).

3. Line 544-45 (quantification of LC3 puncta). Specify number of cells examined.

Response: We have indicated this information in the revised manuscript (page 24, line 552).

4. Lines 165, 172: equal amountS

Response: We have corrected this term in the revised manuscript (page 5-6, lines 168, 180).

5. Line 174-175. Consistent with this.. (consistently means something else); In a control experiment, there was...

Response: We have changed the description in the revised manuscript as recommended by the reviewer (page 6, lines 182-183).

6. Line 224: demonstrating that orf2p HAS

Response: We have corrected this in the revised manuscript (page 7, line 230).

7. Line 241: why "urgently"?

Response: As suggested by the reviewer, we have removed this word from the manuscript (page 7, line 247).

8. Line 247. which LEADS? us to speculate that

Response: We have corrected this in the revised manuscript (page 8, line 262).

Jeffrey Bergelson

Response: Thank you very much. We are deeply grateful for the high-quality review.

Reviewer #3 (Remarks to the Author):

Comment: Most importantly, as also raised by reviewer #2 as specific comment no 1, the outcomes of nearly all experiments are difficult to interpret because they were done at low MOI infection and analysed at 48 to 72 h after infection, implicating that the infection involved 4 multiple rounds of infection. In response to this comment, the authors have performed only one single-cycle experiment, which is shown in Extended Data Figure 7. Here they only analysed virus titers in the supernatants of HT-29 cells. Why did they not include the cellular titers, to show that similar amounts of virus were produced. In addition, why did they not include RD cells which are important controls

Response: As suggested by the reviewer, we have incorporated all the recommended data into revised Figure 4d and Supplementary Figure 7.

Reviewer #3: First, the authors have NOT incorporated ALL the recommended data in Figure 4d. Initially all experiments were performed at low MOI and then the readout was done at 48 post infection which involves multiple rounds of replication. This was a serious omission that was noted by reviewer #2. In response to that question, the authors only come up with one single experiment that was shown in Suppl Fig 7 (HT-29 cells), which was disappointing as it would have been important to have performed single and multicycle experiments throughout the manuscript in all Figures. In response to my question on virus titers, the authors have now completely deleted Supp Fig S7, and come up with revised Fig 4D, which is on IEC cells, and a new Suppl Fig S7, which is on RD cells. They don't show anything for HT-29 cells for reasons that are not explained...

Response: As suggested by the editor and reviewer, we have incorporated the results of intra- and extracellular viral titers in HT-29 cells infected by the EV-A71 and EV-A71 Δ ORF2p viruses at an MOI of 10 (Supplementary Figure 7a). The data further support that ORF2p facilitates the release of EV-A71 in HT-29 cells.

Comment: However, the issues about the low versus high MOI infections remain. For example in Fig 4B, where the authors show virus titers obtained in a transfection experiment. How is it possible that upon RNA transfection the amount of ORF2p-lacking viral RNA in cells is higher than of wt virus at 48 h posttransfection? The ORF2p virus should have a defect in release and spreading and should therefore be less capable in spreading to other cells and should therefore have lower viral RNA levels. Here, as with other low MOI infection studies, the analysis is blurred because the experiment involves many rounds of replication. This example shows how important it is to analyse virus titers after a single round of replication (i.e. high MOI experiments).

Response: We thank the reviewer for pointing out this issue. The suggestion that we assess the virus titre after a single round of replication has greatly strengthened our conclusions. To address the issue raised, we have performed the recommended onestep growth curve experiments for EV-A71 and EV-A71 Δ ORF2p viruses in RD, HT-29, and primary IECs at an MOI of 10 (new Figure 2a, 2b, 2c). The results indicate that ORF2p expression is required for the full replication capacity of EV-A71. In addition, we further confirmed that ectopic expression of ORF2p could restore replication of the ORF2p-defective EV-A71 Δ ORF2p viruses in HT-29 cells at a high MOI (new Figure 2d). Lacking ORF2p expression significantly decreased EV-A71 Δ ORF2p (MOI=10) replication in diverse intestinal cell lines compared to that of EV-A71 wild-type viruses (MOI=10) (Supplementary Figure 3).

Reviewer #3: It is kind that the authors thank me for pointing out this issue, but I cannot understand at all that they did not see this themselves and makes me wonder whether they have full understanding of the design and interpretation of their experiments. It remains a mystery why there is no significant difference in the IEC cells (i.e. the frozen-thawed lysates) infected with wt and ORF2p lacking virus at 48 h posttransfection. In this RNA transfection experiment, virus production also likely involves multiple rounds of replication. As ORF2p-lacking virus release is severely impaired, this should have resulted in reduced levels in the ORF2p-lacking virus infected cell culture. Because no virus titers have been determined at various times posttransfection between 12 and 48 h, this remains a very poor experiment.

Response: As suggested by the reviewer, we have performed the recommended experiments and incorporated them into the revised manuscript (new Figure 4c). The results were consistent with our conclusion.

Similar as my comment on Fig. 4B, also data shown in Figs 4c-e of the previous manuscript suffered from wrong set up. In Fig 4C, the authors found higher virus titers in cells infected with ORF2p lacking virus, which does not make any sense when the virus has a defect in release. Where have these figures gone? They have not been corrected. Instead, the authors have just removed them from the new version without any explanation. It seems that authors prefer to selectively include results that match their ideas, while leaving out results that do not match with their message...

Response: In Figure 4c of the previously submitted manuscript, we measured the levels of viral RNA in the supernatant and cell lysates of EV-A71- or EV-A71 Δ ORF2p-infected HT-29 cells by using the RT-PCR assay. The results showed that vRNA in the supernatant of EV-A71 Δ ORF2p-infected HT-29 cells was severely reduced (44.2-fold) compared to that in the supernatant of EV-A71-infected cells, while the level of intracellular EV-A71 Δ ORF2p vRNA was 1.4-fold higher than the level of intracellular EV-A71 vRNA. This phenomenon could be due to the lack of ORF2p-dependent release of EV-A71 Δ ORF2p RNA into the culture supernatant, which leads to higher rates of intracellular vRNA accumulation compared to that of EV-A71. In addition, we observed cytopathic effects in EV-A71-infected HT-29 cells, which may further decrease intracellular EV-A71 vRNA levels. Similarly, we noticed that there were modestly higher

vRNA levels at 8 and 10 hours post-transfection in the EV-A71 Δ ORF2p RNA-transfected IECs than in the cells transfected with EV-A71 RNA (new Fig. 4b). To address the issue raised by the reviewer, we have acknowledged and discussed the fact that ORF2p-deleted virus accumulates higher RNA levels in the cells in the revised manuscript (page 5, lines 169-173).

The authors have included new Figs 2a-c in their new manuscript. The results shown in 2c (growth in IECs) differ significantly from those in the previous Fig 2c that was included in the previous manuscript. Previously, they found that replication was similar for wt and ORF2p-lacking virus up to 12 h, thereafter there was a difference (which was in line with no role in RNA replication but in virus release). Instead now, they show that virus replication of the ORF2p virus is severely impaired in the ORF2p-lacking virus. This demonstrates very poor reproducibility of the results provided by these authors. Furthermore, the data in the new Fig 2c (where there is a 500-fold difference in virus titers between wt and ORF2p-lacking virus) are inconsistent with those in new Fig 4d, where it is shown that they replicate to similar titers. Overall, the data seem poorly reproducible and inconsistent.

Response: In addition to the possibility that isolating the IECs at different times may have influenced the results (Fig. 2c), we speculate that the difference in our results could partially result from performing the experiments in the revised manuscript at a much higher MOI (10 versus 0.1), which enabled us to detect dramatic differences in the replication capacities of the EV-A71 and EV-A71 Δ ORF2p viruses at 12 hpi in our virus titer assay (Fig. 2a-c). All our experiments were performed in triplicate.

To make their point that ORF2p is important for virus release in intestinal cells, the authors have incorporated more intestinal cell lines in Suppl Fig 3. However, they have replaced the old Suppl Fig 3, which showed that ORF2p is also important for release in NSC-34 cells, which are mouse neural cells (as remarked by Reviewer #1). So, it seems that the authors without any scruples just remove data that do not fit their hypothesis, rather than discussing and softening their conclusions. This is a big disgrace and can mislead the readership!

Response: As suggested by the editor and reviewer, we have put back the original data on NSC-34 cells that was included the previous submission (Supplementary Figure 3b) and incorporated the discussion in the revised manuscript (page 4, lines 119-121).

Comment: The authors failed to adequately respond to comment no 2, where reviewer #2 asks for additional support for a role of ORF2p in particle release. Instead of providing this information, the authors only exclude effects on a number of other steps in the viral life cycle.

Response: We apologize for our poor description. We have performed the experiments and confirmed that ORF2p is required for viral RNA and infectious particle release from EV-A71 RNA transfected IECs (Figure 4a-c). ORF2p deficiency specifically decreased the amount of infectious EV-A71 Δ ORF2p virus particles in the supernatant (Figure 4d,4e). Moreover, ectopic expression of ORF2p restored accumulation of the viral structural protein VP1 in the supernatant of EV-A71 Δ ORF2p-infected cells (Figure 4f).

We have incorporated the required data into the manuscript (Lines 158-179).

Reviewer #3: Okay

Response: Thank you very much.

Comment: The authors also failed to adequately respond to comment no 3, where reviewer #2 asks for more detailed experiment to quantify effects on translation and replication. The authors do not show any experiments to study replication, e.g. with replicons, which is definitely needed and thereby a serious omission.

Response: We greatly appreciate the reviewer for this suggestion. We have generated luciferase replicons for EV-A71 and EV-A71 Δ ORF2p by replacing the P1 structure 5 region with the firefly luciferase gene. The EV-A71 and EV-A71 Δ ORF2p pseudoviruses were packaged in HEK293T cells. RD and HT-29 cells were incubated with equal amounts of pseudoviruses and harvested at 9 h post-infection for luciferase activity assays. The results demonstrate that a lack of ORF2p did not significantly influence the early steps of enterovirus replication (Supplementary Figure 6e).

Reviewer #3: Okay

Response: Thank you very much.

Comment: With regard to comment no 5, the evidence that is provided suggesting that ORF2p localizes to the extracellular vesicles and that it regulates transport and/or release of these vesicles is far from convincing. At best, the authors show some preliminary data that point to such functions (e.g. colocalization with LC3 and the wortmannin experiment, although this latter is not convincing), but experimental evidence that provides firm support for their suggestions is lacking.

Response: We thank the reviewer for this comment. The product of the newly identified ORF of human enteroviruses in the present study, i.e., ORF2p, contributes to full replication capacity in the intestinal cells by facilitating virus particle release. Considering the importance of cell autophagy in enterovirus replication, the identification that ORF2p stimulates autophagy in EV-A71-infected or ORF2p-overexpressing cells should be helpful for a better understanding of the pathogenesis of enteroviruses. The mechanisms of ORF2p-triggered autophagy and its roles in virus replication still need to be further investigated in future studies. We discuss the reviewer's comment in the revised manuscript (Page 7, lines 230-242).

Reviewer #3: The authors spend a few lines of text, but still I am not convinced by their data. The data remain superficial. Unfortunately, no new attempts to provide stronger data and to convince the readership have been undertaken.

Response: We appreciate the reviewer for pointing out this issue. In the present study, our findings challenge the long-held notion that the enterovirus genome contains only a single ORF that encodes a single polyprotein and show that HEVs harbour a second ORF that encodes the ORF2p protein to support viral intestinal infection. Cell autophagy

has been broadly accepted to be essential for enterovirus replication *in vitro* and *in vivo*, and in Supplementary Figure 9, we showed that ORF2p is a novel inducer of autophagic flux in EV-A71-infected or ORF2p-overexpressing intestinal cells, which may contribute to efficient viral replication. The mechanisms by which ORF2p induces intracellular autophagy responses and the effects of ORF2p-mediated autophagy on the pathogenesis of HEVs remain unknown and should be investigated in future studies. In the revised manuscript, we have added the necessary discussion to strengthen our conclusion (page 6, lines 210-214; page 7, lines 245-247).